# Albanol B from Mulberries Exerts Anti-Cancer Effect through Mitochondria ROS Production in Lung Cancer Cells and Suppresses In Vivo Tumor Growth

**DOI:** 10.3390/ijms21249502

**Published:** 2020-12-14

**Authors:** Thanh Nam Phan, Okwha Kim, Manh Tuan Ha, Cheol Hwangbo, Byung-Sun Min, Jeong-Hyung Lee

**Affiliations:** 1Department of Biochemistry, College of Natural Sciences, Kangwon National University, Chuncheon, Gangwon-Do 24414, Korea; thanhnam1719@gmail.com (T.N.P.); kah3173@kangwon.ac.kr (O.K.); 2College of Pharmacy, Catholic University of Daegu, Gyeongbuk 38430, Korea; hamanhtuan238@gmail.com (M.T.H.); bsmin@cu.ac.kr (B.-S.M.); 3Division of Applied Life Science (BK21 Plus), Plant Molecular Biology and Biotechnology Research Center, Gyeongsang National University, Jinju 52828, Korea; chwangbo@gnu.ac.kr; 4Division of Life Science, College of Natural Sciences, Gyeongsang National University, Jinju 52828, Korea

**Keywords:** Albanol B, apoptosis, cell cycle arrest, mitochondrial reactive oxygen species, lung cancer

## Abstract

Albanol B (ABN-B), an arylbenzofuran derivative isolated from mulberries, has been shown to have anti-Alzheimer’s disease, anti-bacterial and antioxidant activities. The aim of this study was to investigate the anti-cancer effect of this compound against lung cancer cells. The results show that ABN-B inhibited the proliferation of four human lung cancer cell lines (A549, BZR, H1975, and H226) and induced apoptosis, based on the cleavage of caspase-7 and PARP (poly (ADP-ribose) polymerase), as well as the downregulation of Bcl-2. ABN-B also induced cell cycle arrest at G_2_/M by down-regulating the expression of CKD1 (cyclin-dependent kinase 1) and cyclin B1, but up-regulating p21 (cyclin-dependent kinase inhibitor 1) expression. Notably, ABN-B increased the production of mitochondrial reactive oxygen species (ROS); however, treatment with mito-TEMPO (a specific mitochondrial antioxidant) blocked ABN-B-induced cell cycle arrest at G_2_/M and apoptosis, as well as the up-regulation of p21 and down-regulation of CDK1 and cyclin B1 induced by ABN-B. At the molecular level, ABN-B-induced mitochondrial ROS production increased the phosphorylation levels of AKT (protein kinase B) and ERK1/2 (extracellular signal-regulated kinase 1/2), while the inhibition of these kinases blocked the ABN-B-induced up-regulation of p21 and down-regulation of CDK1 and cyclin B1. Moreover, ABN-B significantly suppressed tumor growth in Ex-3LL (Lewis lung carcinoma) tumor-bearing mice. Taken together, these results suggest that ABN-B can exert an anti-cancer effect by inducing apoptosis and cell cycle arrest at G_2_/M through mitochondrial ROS production in lung cancer cells.

## 1. Introduction

Lung cancer is the most commonly diagnosed cancer among all malignancies in the world and the leading cause of cancer death [1]. Over 2.2 million cases of lung cancer are diagnosed in 2018, accounting for 11.6% of the total cases, and 1.76 million deaths in 2018 are attributed to lung cancer, accounting for 18.4% of all cancer-related deaths [1]. The most common type of lung cancer is adenocarcinoma, comprising around 40% of all lung cancer cases [2]. Despite the advances in understanding this disease mechanism and the development of new therapeutic strategies, lung adenocarcinoma is still one of the most aggressive and rapidly fatal types of cancer [3,4]. Thus, new therapeutic agents with high efficiency for this deadly disease are urgently needed.

Reactive oxygen species (ROS) are involved in the regulation of many important cellular processes such as cell cycle and apoptosis [5,6,7]. For example, increased ROS generation in cancer cells activates signaling pathways necessary for the initiation, promotion and progression of tumors, and also contributes to tumor resistance to chemotherapy [8]. ROS can activate certain specific signaling pathways, including hypoxia-inducible factors (HIFs), phosphatidylinositol 3-kinase (PI3K)/AKT and mitogen-activated protein kinase (MAPKs) cascade kinases [5,9,10,11]. However, the extremely reactive hydroxyl radical form of ROS generated from hydrogen peroxide can oxidize proteins, lipids and DNA massively, leading to damage or genomic instability [12,13]. Cancer cells exhibit higher ROS levels than normal cells [8]. However, cancer cells have an increased activity of antioxidant enzymes to counteract these ROS levels. Thus, therapeutic strategies that either increase ROS generation and/or decrease antioxidant defense in cancer cells may lead to the activation of various cell death pathways and thus limit the cancer progression [7,8].

*Morus alba*, commonly known as white mulberry, belongs to the Moraceae family. In Asian countries, the leaves, fruits and barks of *M. alba* have long been used as traditional medicine to protect liver damage, improve eyesight and lower blood pressure [14]. It possesses many biological activities, including anti-hyperlipidemic, anti-hypertensive, anti-hyperglycemic, anti-microbial, anti-allergic, anti-inflammatory, hepatoprotective and neuroprotective activities [10,11]. Phytochemical studies of *M. alba* have revealed various constituents, including terpenoids, alkaloids, flavonoids, phenolic acids, stilbenoids and coumarins [14,15]. These constituents exert a variety of pharmacological effects, including anti-diabetes [16], antioxidant [17,18], anti-inflammatory [19,20], neuroprotective [21] and anticancer effects [22]. For example, morusin, a prenylated flavonoid isolated from the root bark of *M. alba*, has been shown to inhibit cell proliferation and tumor growth by downregulating c-Myc oncogene in human gastric cancer [23], and has also been shown to induce apoptosis in human non-small cell lung cancer cells by suppressing EGFR/STAT activation [24].

Albanol B (ABN-B) is an arylbenzofuran derivative isolated from *M. alba*. ABN-B and has been demonstrated to possess antioxidant [25], anti-inflammatory [26], anti-Alzheimer [27] and anti-diabetes properties [28,29]. However, further study is needed to elucidate the anti-cancer effect of this compound and its underlying mechanism. As part our continuing search for an anti-cancer compound from *M. alba,* we investigated the anti-cancer activity of ABN-B in lung cancer cell lines. As a result, we found that ABN-B exhibited in vitro and in vivo anti-cancer activity that triggered cell cycle arrest at G_2_/M and apoptosis. Moreover, we demonstrated that ABN-B showed active anti-cancer activity by enhancing the mitochondrial ROS production in human lung cancer cells.

## 2. Results

### 2.1. ABN-B Inhibits the Proliferation of Human Lung Cancer Cells

To evaluate the anti-cancer activity of ABN-B, we performed MTT assays on four human lung cancer cell lines: A549, BZR, NCI-H1975 and NCI-H226. The results showed that treatment with ABN-B for 24 h significantly reduced the cell viability of A549 cells at concentrations of 10 and 30 µM; however, ABN-B did not significantly decrease the cell viability of other cells lines at concentrations of 1, 3, and 10 µM (Figure 1A). Treatment with ABN-B for 48 h resulted in a significant concentration-dependent decrease in the cell viability of these lung cancer cell lines (Figure 1B). The IC_50_ values of ABN-B for 48 h treatment against A549, BZR, NCI-H1975 and NCI-H226 cells were 5.6 ± 0.4, 8.9 ± 0.6, 12.7 ± 1.0, and 15.0 ± 3.3 μM, respectively, while the respective IC_50_ values of etoposide under the same condition were 23.5 ± 2.8, 15.8 ± 1.4, 18.5 ± 2.1 and 14.4 ± 1.9 μM (Appendix A). Non-small cell lung cancer (NSCLC) comprises approximately 80–85% of all lung cancers [30]. Thus, we selected two NSCLC cancer cell lines, H1975 and A549 cells, for further investigation of the anticancer effect of ABN-B. Bromodeoxyurine (BrdU) incorporation and colony formation assays revealed that ABN-B was also shown to inhibit the proliferation and colony formation of A549 and NCI-H1975 cells in a concentration-dependent manner (Figure 1C,D).

### 2.2. ABN-B Induces Cell Cycle Arrest at G2/M in A549 and NCI-H1975 Cells by Down-Regulating CDK1 and Cyclin B1, but Up-Regulating p21 Protein Expression

Next, we examined whether the growth inhibitory effect of ABN-B is associated with an alteration in the cell-cycle progression of lung cancer cells, and confirmed that ABN-B induced a prominent G_2_/M arrest in the cell-cycle progression of A549 and NCI-H1975 cells after 48 h treatment (Figure 2A,B). Compared with vehicle treated control cells, ABN-B treatment increased G_2_/M phase and decreased G_0_/G_1_ phase in concentration-dependent manners. The treatment of A549 with 3, 10 and 30 μM ABN-B for 48 h increased cells in G_2_/M phase from 9.5% to 14.34%, 18.13% and 47.9%, respectively (Figure 2A). The treatment of NCI-H1975 with 10 and 30 μM ABN-B for 48 h also increased cells in G_2_/M phase from 12.1% to 21.11% and 33.9%, respectively (Figure 2C).

Since A549 cells were more sensitive to ABN-B for inhibiting BrdU incorporation (Figure 1C) and inducing cell cycle arrest (Figure 2A,B) than H1975 cells, we chose this cell line for further studies. To investigate the mechanisms by which ABN-B induced G_2_/M phase cell cycle arrest, we analyzed the effect of ABN-B on the expression of cell-cycle regulatory proteins which could be involved in the G_2_/M arrest. Western blot analysis showed that ABN-B up-regulated the expression of CDK inhibitor p21^Waf1^, but down-regulated the expressions of cyclin B1 and CDK1 in concentration- and time-dependent manners (Figure 2C,D). These results indicate that ABN-B could induce cell cycle arrest at G_2_/M in human lung cancer cells, possibly through the combined down-regulation of cyclin B1 and CDK1 and up-regulation of p21^Waf1^.

### 2.3. ABN-B Induces Apoptosis in A549 Cells

To further examine whether ABN-B exerts an anti-proliferative effect by inducing apoptosis in human cancer cell lines, we evaluated its apoptosis-inducing activities using Annexin-V/PI double staining. The treatment of A549 cells with ABN-B increased the populations of Annexin-V^+^/PI^+^ (late apoptosis) and Annexin-V^+^/PI^−^ early apoptosis) cells in concentration- and time-dependent manners, suggesting that ABN-B induced apoptosis (Figure 3A,B). We also investigated whether ABN-B increases the expression levels of apoptotic proteins through Western blotting (Figure 3C). The treatment of A549 cells with ABN-B increased the levels of cleaved PARP and caspase-7, but decreased the expression level of anti-apoptotic Bcl-2 protein in a concentration-dependent manner (Figure 3C), indicating that this compound may have the ability to induce apoptosis in human lung cancer cells.

### 2.4. ABN-B Induced Mitochondrial ROS Production in Both A549 and H1975 Cells

Due to the critical role of ROS in the regulation of cell proliferation and survival [6,7], we examined whether ABN-B induces ROS production in human lung cancer cells. To this end, we measured cellular and mitochondrial ROS using DCF-DA and Mito-SOX, respectively. The treatment of A549 cells with ABN-B increased both DCF-DA and Mito-SOX fluorescence in concentration- and time-dependent manners (Figure 4A–D). Similar results were also observed in H1975 cells (Figure 4E–H). However, co-treatment with either NAC, an antioxidant, or Mito-TEMPO, a specific mitochondrial ROS scavenger, significantly decreased ABN-B-induced ROS production. These results indicate that ABN-B could induce mitochondrial ROS production in human lung cancer cells.

### 2.5. Mito-TEMPO Attenuated ABN-B-Induced Apoptosis and G2/M Phase Cell Cycle Arrest in Human Lung Cancer Cells

We next examined whether Mito-TEMPO could inhibit ABN-B-induced cell cycle arrest and apoptosis in human lung cancer cells. The treatment of A549 cells with Mito-TEMPO significantly attenuated ABN-B-induced cell cycle arrest at G_2_/M phase from 40.2% to 27.2% (Figure 5A). Similarly, Mito-TEMPO also inhibited ABN-B-induced cell cycle arrest at G_2_/M phase from 49.8% to 16.8% in NCI-H1975 cells (Figure 5B). We further confirmed these results by determining the effect of Mito-TEMPO on the expression levels of cyclin B1, CDK1 and p21^Waf1^ induced by ABN-B. The treatment of A549 cells with Mito-TEMPO significantly reversed both the ABN-B-induced up-regulation of p21 and the down-regulations of cyclin B1 and CDK1 (Figure 5C), suggesting that ABN-B could induce cell cycle arrest at the G_2_/M phase in human lung cancer cells via mitochondrial ROS generation. We also conducted Annexin V-FITC/PI double staining to determine whether Mito-TEMPO attenuates ABN-B-induced apoptosis. The treatment of A549 cells with Mito-TEMPO was shown to significantly prevent the apoptosis and the cleavage of PARP, as well as the down-regulation of Bcl-2 expression induced by ABN-B (Figure 5D,E), suggesting that ABN-B may trigger apoptosis in A549 cells via mitochondrial ROS generation. Altogether, these results demonstrate the critical role of mitochondrial ROS in the cell cycle arrest and apoptosis induced by ABN-B in human lung cancer cells.

### 2.6. ABN-B Induced the Phosphorylation of ERK1/2 and AKT via the Mitochondrial ROS Production

We investigated the signaling pathways by which ABN-B induced cell cycle arrest and apoptosis in A549 cells. The treatment of A549 cells with ABN-B increased the phosphorylation levels of ERK1/2 and AKT in time- and concentration-dependent manners (Figure 6A,B). However, ABN-B did not increase the phosphorylation level of JNK and p38 MAPKs (Figure 6A,B). Co-treatment with Mito-TEMPO significantly reduced the phosphorylation levels of ERK-1/2 and AKT induced by ABN-B (Figure 6C). To further confirm that the mitochondrial ROS production induced by ABN-B is an upstream event of the activation of ERK1/2 and AKT, we determined the effect of U0126 (MEK inhibitor) and LY294002 (PI3K inhibitor) on ABN-B-induced mitochondrial ROS production (Figure 6D). The results showed that co-treatment with U0126 and LY294002 did not affect ABN-B-induced mitochondrial ROS production. Taken together, these results suggest that ABN-B-induced mitochondrial ROS production could mediate the activation of ERK-1/2 and AKT in A549 cells.

### 2.7. Inhibition of ERK1/2 and AKT Attenuated ABN-B-Induced Apoptosis and G2/M Arrest

We next assessed the effects of U0126 and LY294002 on ABN-induced apoptosis and cell cycle arrest at G_2_/M in A549 cells. Co-treatment with U0126 or LY294002 blocked the ABN-B-induced down-regulation of Bcl-2 and cleavage of PARP (Figure 7A,B), suggesting that the activation of ERK1/2 and AKT could be critical aspects of ABN-B-induced apoptosis in A549 cells. Interestingly, co-treatment with U0126 blocked the ABN-B-induced down-regulation of cyclin B1, but not that of CDK1. By contrast, co-treatment with LY294002 blocked the ABN-B-induced down-regulation of CDK1, but not that of cyclin B1 (Figure 7A,B), suggesting that the ERK1/2 and AKT pathways could play critical roles in the ABN-B-induced down-regulations of cyclin B1 and CDK1, respectively.

### 2.8. ABN-B Suppressed Tumor Growth in Ex-3LL Tumor-Bearing Mice

To determine whether ABN-B suppresses the growth of lung cancer cells in vivo, we evaluated the effect of ABN-B on the growth of EX-3LL cells in a syngeneic mouse model. The results showed that ABN-B significantly suppressed tumor growth of EX-3LL tumor-bearing mice (Figure 8A). The relative inhibition rate of tumor volume and weight (Figure 8B) decreased by 48.0% and 41.0% (for 50 mg/kg treatment), and 61.3% and 57.3% (for 100 mg/kg treatment), respectively, after treatment with ABN-B for 21 days in EX-3LL tumor-bearing mice. On the other hand, no significant differences in body weights were found between vehicle- and ABN-B-treated group (Figure 8D). We also determined whether ABN-B induced apoptosis in Ex-3LL tumor tissue by terminal deoxynucleotidyl transferase-mediated dUTP nick-end labeling (TUNEL) based staining. The TUNEL-positive cells were not detected in tumor tissues of the vehicle-treated group. However, we observed numerous TUNEL-positive cells in tumor tissues of ABN-B-treated group (Figure 8E). These results indicate that ABN-B could exert potent anti-cancer effects in vivo in lung cancer cells.

## 3. Discussion

Despite the substantial progress that has been made in the treatment of lung cancer, lung cancer remains the most lethal type of cancer worldwide, accounting for more than 1.7 million deaths each year [1]. In the present study, we found that ABN-B, an arylbenzofuran derivative from *M. alba*, exerts an anti-cancer effect in vitro and in vivo models of lung cancer cells by inducing cell cycle arrest at G_2_/M and apoptosis in lung cancer cells. Mechanistically, ABN-B leads to the activation of the PI3K/AKT and ERK1/2 MAPK pathways via the production of mitochondrial ROS, which resulted in the induction of apoptosis and cell cycle arrest at G_2_/M in human lung cancer cells (Figure 8E). To our knowledge, this is the first report to demonstrate the anti-cancer effect in vitro and in vivo of ABN-B in lung cancer cells.

Natural products have long and rich experience in treatment of cancer and are accepted increasingly as complementary and alternative therapy [31]. Anticancer drugs discovered from natural products have been used clinically for cancer treatment as the conventional anticancer drugs [30]. Extract of *M. alba* has been shown to exert anticancer effects on various human cancer cell lines. For example, extract of *M. alba* root bark induces cell growth arrest and apoptosis in SW480 human colorectal cancer cells by activating ATF3 expression and down-regulating cyclin D1 level [32], and also reduces the viability of multidrug-resistant MCF-7/Dox cells by inhibiting YB-1-dependent MDR1 expression [33]. Furthermore, many compounds with a variety of antitumor mechanisms have been identified from *M. alba*. For example, one study has indicated that mulberrofurans, moracins, sanggenon O and albafuran A inhibit hypoxia-inducible factor-1 accumulation and hypoxia-induced vascular endothelial growth factor secretion in Hep3B cells [34]. Mulberrofuran G has been shown to induce apoptotic cell death in HL-60 cells via both the cell death receptor pathway and the mitochondrial pathway [35]. In the present study, we showed that ABN-B exerted an anti-proliferative effect on human lung cancer cells by inducing cell cycle arrest at G2/M and apoptosis. Moreover, ABN-B dose-dependently suppressed Ex-3LL tumor growth in a syngeneic implantation model, indicating that ABN-B could be considered as a new lead compound for the development of anti-cancer agent against lung cancer. Moreover, extract of *M. alba* could be of value for further exploration as a potential anti-cancer agent for the treatment of lung cancer.

ROS plays a dual role in cancer [12]. On the one hand, ROS can promote protumorigenic signaling, facilitating the proliferation, survival and adaptation to hypoxia of cancer cells. On the other hand, ROS can promote anti-tumorigenic signaling and trigger oxidative stress–induced cancer cell death. Cancer cells have an increased ROS level compared with normal cells due to their high metabolic rate and mitochondrial dysfunction, which leads to an increased susceptibility to oxidative stress [8]. Thus, ROS can eventually increase beyond a certain threshold level that is incompatible with cellular survival, resulting in oxidative stress-induced cell death. Several natural products that increase cellular ROS levels have been shown to selectively target cancer cells [36]. For example, dietary phytochemicals such as polyphenols, flavonoids, and stilbenes have the capacity to inhibit cancer cell proliferation and induce apoptosis and autophagy [36,37]. For example, piperine, the most abundant alkaloid found in *Piper longum*, suppresses tumor growth in vitro and in vivo by inducing cell cycle arrest at G_2_/M and apoptosis via ROS production [38,39]. In the present study, we demonstrated that ABN-B increased the productions of both intracellular and mitochondrial ROS, and that co-treatment with mito-TEMPO, a specific scavenger of mitochondrial ROS, suppressed ABN-induced cell cycle arrest at G_2_/M and apoptosis, indicating that ABN-B has an anti-proliferative effect in human lung cancer cells by inducing mitochondrial ROS production.

Research has shown that ERK activity can promote apoptotic pathways, cell cycle arrest or autophagic vacuolization [40]. These effects require sustained ERK activity in specific subcellular compartments and could depend on the presence of reactive oxygen species [40]. Studies have shown that ROS promotes sustained ERK activation by promoting the activation of tyrosine kinase receptors and by inhibiting ERK-directed phosphatases such as DUSP1 and DUSP6 [41,42,43,44]. The AKT signaling pathway has also been implicated in sensitizing cells to apoptosis, and the ROS-mediated activation of AKT induces apoptosis in prostate cancer cells [45,46]. For example, piroxicam, a traditional non-steroidal anti-inflammatory drug, causes apoptosis by ROS-mediated AKT activation, and artocarpin, an isoprenyl flavonoid, induces apoptosis via the ROS-mediated activation of MAPKs and AKT in non-small cell lung cancer cells [47,48]. In the present study, we demonstrated that ABN-B induced the phosphorylation of ERK1/2 and AKT in human lung cancer cells, but not of JNK and p38 MAPKs, via mitochondrial ROS production. These two pathways appeared to be involved in both ABN-B-induced cell cycle arrest and apoptosis. Co-treatment of U0126, a specific MEK inhibitor, inhibited the ABN-B-induced up-regulation of PARP cleavage as well as the down-regulation of Bcl-2 and cyclin B1 expressions, but not of CDK1. Co-treatment of LY294002, a specific PI3K inhibitor, also inhibited the ABN-B-induced up-regulation of PARP cleavage and down-regulation of Bcl-2, suggesting that both of the ERK-1/2 and AKT pathways might be involved in the induction of ABN-B-mediated apoptosis. However, LY294002 inhibited the ABN-B-induced down-regulation of CDK1 expression, but not that of cyclin B1, suggesting that the ERK and AKT pathways could regulate the ABN-induced down-regulation of cyclin B1 and CDK1 expressions, respectively.

In summary, the present study demonstrated for the first time that ABN-B, an arylbezofuran derivative from *M. alba*, exerts anti-cancer in vitro and in vivo in lung cancer cells which are associated with mitochondrial ROS production and the subsequent activation of ERK and AKT, suggesting that the ABN-B-induced mitochondrial ROS production could be an important mechanism for its anti-cancer effect. These study findings provide a rational basis for the usage of *M. alba* extracts for cancer treatment in traditional oriental medicine. In addition, ABN-B might be a valuable compound meriting further research as an anti-cancer agent. However, the precise mechanisms by which ABN-B increases mitochondrial ROS production must be further elucidated to better understand the anti-cancer activity of ABN-B.

## 4. Materials and Methods

### 4.1. Cell Culture

A549, BZR, H1975 and H226 cell lines were purchased from the American Type Culture Collection (Manassas, VA, USA). A549, H1975 and H226 cells were maintained in RPMI 1640 medium, while BZR cells were maintained in DMEM. A murine Ex-3LL (Lewis lung carcinoma) cell line, which is derived from 3LL, was purchased from JCRB Cell Bank (Osaka, Japan) and maintained in RPMI 1640 medium. All media were supplemented with penicillin-streptomycin (Invitrogen, Carlsbad, CA, USA) and 10% heat-inactivated FBS (Hyclone, Logan, UT, USA) and all cells were cultured in a humidified chamber with a 5% CO_2_ atmosphere at 37 °C.

### 4.2. Chemicals and Reagents

Anti-caspase-7, anti-Akt, anti-phospho-Akt (S473), anti-p38, anti-phospho-p38, anti-phospho-ERK1/2, anti-ERK1/2, anti-PARP (Poly [ADP-Ribose] Polymerase), anti-JNK (c-Jun N-terminal kinase) and anti-phospho-JNK antibodies were purchased from Cell Signaling Technology (Danvers, MA, USA). Anti-CDK1, anti-Bcl-2, anti-Cyclin B1, and anti-p21^Waf1^ antibodies were purchased from Santa Cruz Biotechnology (Dallas, TX, USA). MTT (3-(4,5-dimethylthiazol-2-yl)-2,5-diphenyltetrazolium bromide), NAC (*N*-acetyl cysteine), and anti-α-tubulin antibody were purchased from Sigma-Aldrich (St**.** Louis**,** MO, USA)**.** LY294002 and U0126 were obtained from Calbiochem (San Diego, CA, USA). Mito-TEMPO was purchased from Santa Cruz Biotechnology (Dallas, TX, USA).

### 4.3. Isolation of ABN-B

ABN-B was obtained from the twigs of *M. alba*. Briefly, the air-dried powdered root bark of M. alba (10 kg) was extracted with methanol under reflux and then filtered. The methanol extract was concentrated under reduced pressure to give a residue (1.2 kg), which was suspended in distilled water and successively partitioned with n-hexane, dichloromethane and ethyl acetate. The ethyl acetate fraction (304.3 g) was fractionated by extensive column chromatography (CC) with silica gel eluted by a gradient of 0→100% methanol in dichloromethane to afford twenty fractions (F1–F20). Fraction F5 (6.8 g) was fractionated on silica gel CC eluted with dichloromethane-methanol (20:1 to 0:1, *v*/*v*) to afford 16 sub-fractions (F5.1–F5.16). Sub-fraction F5.15 (623 mg) was passed over an RP-C_18_ silica gel column using methanol-water system (2:1, *v*/*v*) as a mobile phase to yield ABN-B (150 mg). The structure and purity of ABN-B were confirmed by ^1^H and ^13^C nuclear magnetic resonance spectra (Appendix A).

### 4.4. Cell Viability and Proliferation Assays

MTT-based colorimetric assay was used to measure the cytotoxic effect of ABN-B. In brief, cells were seeded in 96-well plates (3 × 10^4^ cells per well) and allowed to grow to the plate for 24 h. ABN-B was then treated to the wells at different concentrations and the plates were further incubated for 24 h or 48 h. At the end of incubation, 20 µL of MTT solution (5 mg/mL) was added to each well and was further incubated for 4 h. The concentrations needed to reduce the cell density by 50% (IC_50_ values) were calculated through non-linear regression analysis. Cell proliferation was measured using the BrdU Cell Proliferation Assay Kit (Cell Signaling Technology, Danvers, MA, USA) and performed following the manufacturer’s instructions.

### 4.5. Colony Formation Assay

Colony formation assay was performed as described previously [49]. Briefly, 500 cells per well were seeded in 12-well plates and allowed to adhere to the plate for 24 h, and then ABN-B was added to the wells at different concentrations. Following 48 h incubation, the medium was replaced with fresh medium with or without ABN-B, and the cells were further grown for an additional five days. After the cells were washed with PBS (pH 7.4) twice, the cells were stained with crystal violet solution (0.1%) diluted in ethanol 40% for 5 min. Then, the stain solution was removed using tap water and the cells were air-dried and finally, the number of colonies was counted.

### 4.6. Annexin V/PI Double Staining

The quantification of cell death was evaluated by flow cytometry using an Annexin V-FITC apoptosis detection kit according to the manufacturer’s instructions (BD Biosciences, San Jose, CA, USA). The cells were treated with the indicated concentrations of compounds for either 24 h or 48 h, then were collected by centrifugation. Cell pellet was stained with Annexin V-FITC and propidium iodide (PI) in a binding buffer for 15 min at room temperature in the dark. Annexin V-FITC/PI stained cells were analyzed by flow cytometry (FACS Calibur, Becton-Dickinson, San Jose, CA, USA).

### 4.7. Western Blot Analysis

For the preparation of whole cell lysates, the cells were lysed in a lysis buffer (50 mM Tris-HCl [pH 7.5], 1% Nonidet P-40, 1 mM EDTA, and 150 mM NaCl) in the presence of protease inhibitor cocktail (Sigma-Aldrich) and 5 mM sodium orthovanadate. The lysates were resolved by sodium dodecyl sulfate-polyacrylamide gel electrophoresis and subjected to Western blot analysis. Western blots were incubated overnight with the indicated antibodies (1:1000 dilution) and anti-rabbit or anti-mouse secondary antibodies conjugated to horseradish peroxidase (1:2000 dilution) were used to visualize signals using an enhanced chemiluminescence system (ThermoFisher Scientific, Rockford, IL, USA), and the band intensity was quantified using Image J software (NIH, Bethesda, MD, USA).

### 4.8. Cell Cycle Distribution Analysis

For the determination of cell-cycle distribution, the cells were treated with the indicated concentrations of ABN-B for 48 h, and then washed and centrifuged. The pellets were fixed in ice-cold 80% (*v*/*v*) ethanol for 1 h at 4 °C. The cells were subsequently centrifuged and incubated with cold PI solution (50 µg/mL PI and 0.1 mg/mL RNase A for 30 min in the dark. Cell cycle distribution were analyzed using FACSCalibur (Becton Dickinson, San Jose, CA, USA).

### 4.9. Measurement of Intracellular and Mitochondrial ROS Level

2′,7′-Dichlorofluorescin diacetate (DCF-DA, Sigma-Aldrich) and MitoSOX red (Molecular Probes, Invitrogen, Carlsbad, CA, USA) were used to measure the levels of intracellular ROS and mitochondrial ROS, respectively, as described previously [50]. Cells were pretreated with NAC (100 µM) or Mito-TEMPO (50 µM) for 30 min, followed by the treatment with either the indicated concentrations of ABN-B for 2 h or 30 µM of ABN-B for the indicated period of time. The cells were then treated with either DCF-DA (10 µM) or MitoSOX (5 µM) in the dark for 30 min, and harvested after washing with PBS. Their fluorescence intensities were assessed using a FACS Calibur flow cytometer (Becton-Dickinson, for DCF-DA, excitation wavelength: 485 nm, emission wavelength: 535 nm; for MitoSOX, excitation wavelength: 510 nm, emission wavelength: 580 nm).

### 4.10. In Vivo Tumor Growth Assay and Detection of Apoptosis in Tumor Tissues

All experimental protocols were approved by the Institutional Animal Care and Use Committee (IACUC) of Kangwon National University (IACUC approval No. KW-190403-1, approval date 9 April 2019). For in vivo tumor growth assay, Ex-3LL cells (1 × 10^7^ cells per mouse) were subcutaneously injected into the right flank of 4-week-old male C57BL6 mice (Orient Bio Inc., Seongnam-si, Korea). Tumor growth was monitored by caliper measurements and tumor volume was calculated as the following formula: W^2^ × L × 0.5, where W is the short diameter of the tumor, L is the long diameter of the tumor. Treatment was started when tumor volumes reached approximately 100 mm^3^. Mice (5 mice/group) were injected intraperitoneally with ABN-B (50 or 100 mg/kg) dissolved in dimethyl sulfoxide:chremophore-EL:PBS (1:1:8 by volume) or control vehicle three times a week. The mice were sacrificed after 21 days, and tumors were removed and embedded in OCT compound. Tissue samples were sectioned at a thickness of 5 μm. For antigen retrieval, the slides were pretreated by heating at 95 °C for 5 min in a retrieval buffer (10 mM Tris-HCl, 1 mM EDTA, 0.05% Tween 20, pH 8.0). Apoptosis in tumor tissues was detected using TdT DAB In situ Apoptosis Detection Kit (Trevigen, Gaithersburg, MD, USA).

### 4.11. Statistical Analysis

The results are presented as the mean ± standard error of the mean (SEM). Statistical significance was conducted using one-way analysis of variance (ANOVA) and the differences between the experimental groups were further compared by Fisher’s least significant difference test.

## Figures and Tables

**Figure 1 ijms-21-09502-f001:**
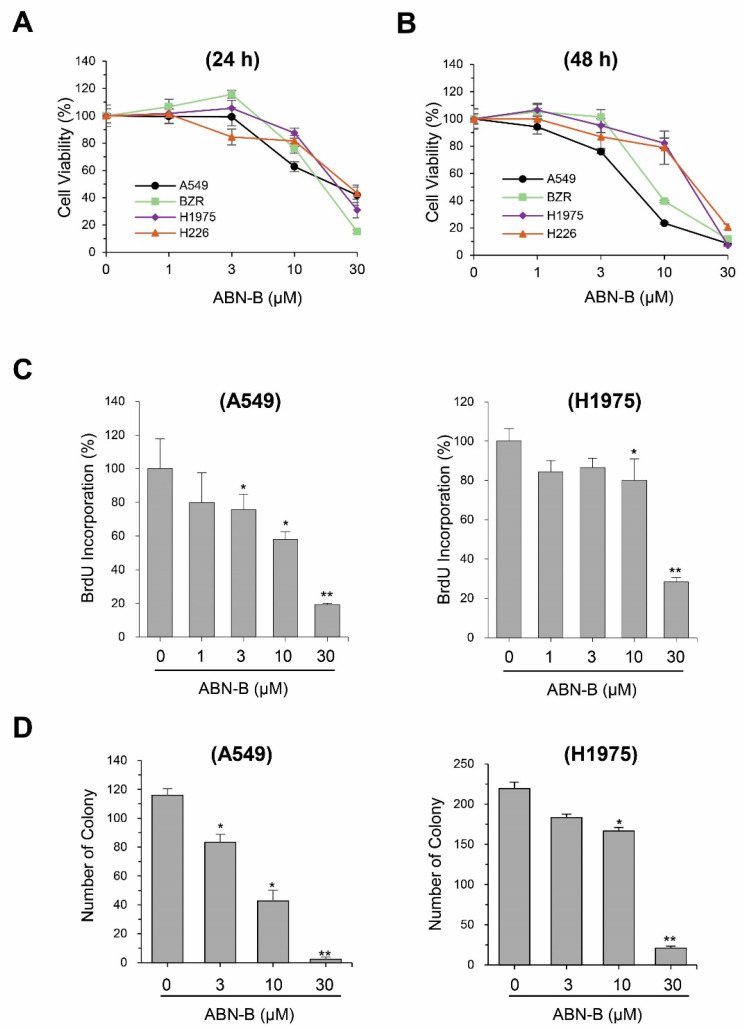
ABN-B inhibited the proliferation of human lung cancer cell lines. (**A**,**B**) A549, BZR, H1975 and H226 cells were treated with the indicated concentrations of ABN-B for 24 h (**A**) or 48 h (**B**). Cells viability was assessed by MTT assay. Data are presented as the mean ± SEM (* *p* < 0.05 and * *p* < 0.01 compared with vehicle-treated control; *n* = 3). (**C**) A549 and NCI-H1975 cells were treated with the indicated concentrations of ABN-B for 24 h and BrdU incorporation was determined. Data are the mean ± SEM (* *p* < 0.05 and ** *p* < 0.01 compared with vehicle-treated control; *n* = 3). (**D**) Colony formation assay of A549 and NCI-H1299 cells was performed in the presence of the indicated concentrations of ABN-B for 6 days. The number of colonies was determined. Data are the mean ± SEM (* *p* < 0.05 and ** *p* < 0.01 compared with vehicle-treated control; *n* = 3).

**Figure 2 ijms-21-09502-f002:**
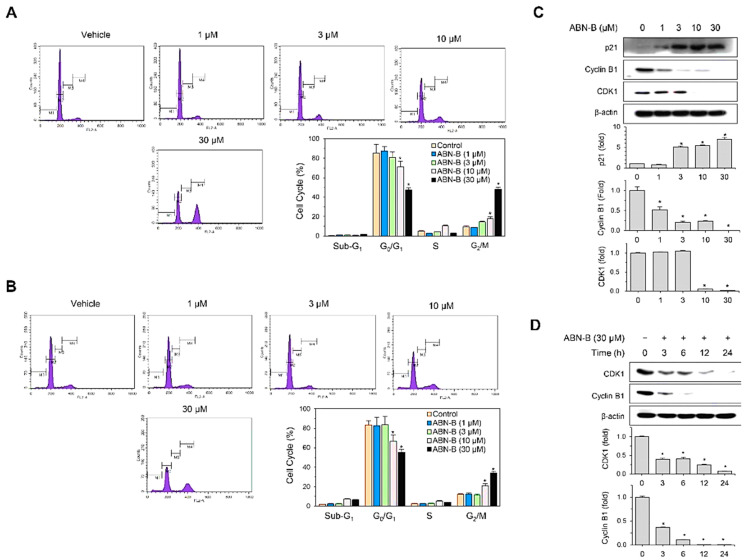
ABN-B induced cell cycle arrest at G_2_/M in A549 and H1975 cells. (**A**,**B**) A549 (**A**) and H1975 (**B**) cells were treated with the indicated concentrations of ABN-B for 48 h, and subsequently stained with PI, followed by analysis using a flow cytometry. Data are expressed as the mean ± SEM of three independent experiments (* *p* < 0.01 compared with vehicle-treated control). (**C**) A549 cells were treated with the indicated concentrations of ABN-B for 48 h. Subsequently, whole cell lysates were prepared, and performed Western blot analysis with the indicated antibodies. Graphs representing densitometry analyses (* *p* < 0.01 compared with vehicle treated control, *n =* 3). (**D**) A549 cells were treated with ABN-B (30 µM) for the indicated periods of time. Subsequently, whole cell lysates were prepared, and performed Western blot analysis with the indicated antibodies. Graphs represent densitometry analyses (* *p* < 0.01 compared with vehicle treated control, *n =* 3).

**Figure 3 ijms-21-09502-f003:**
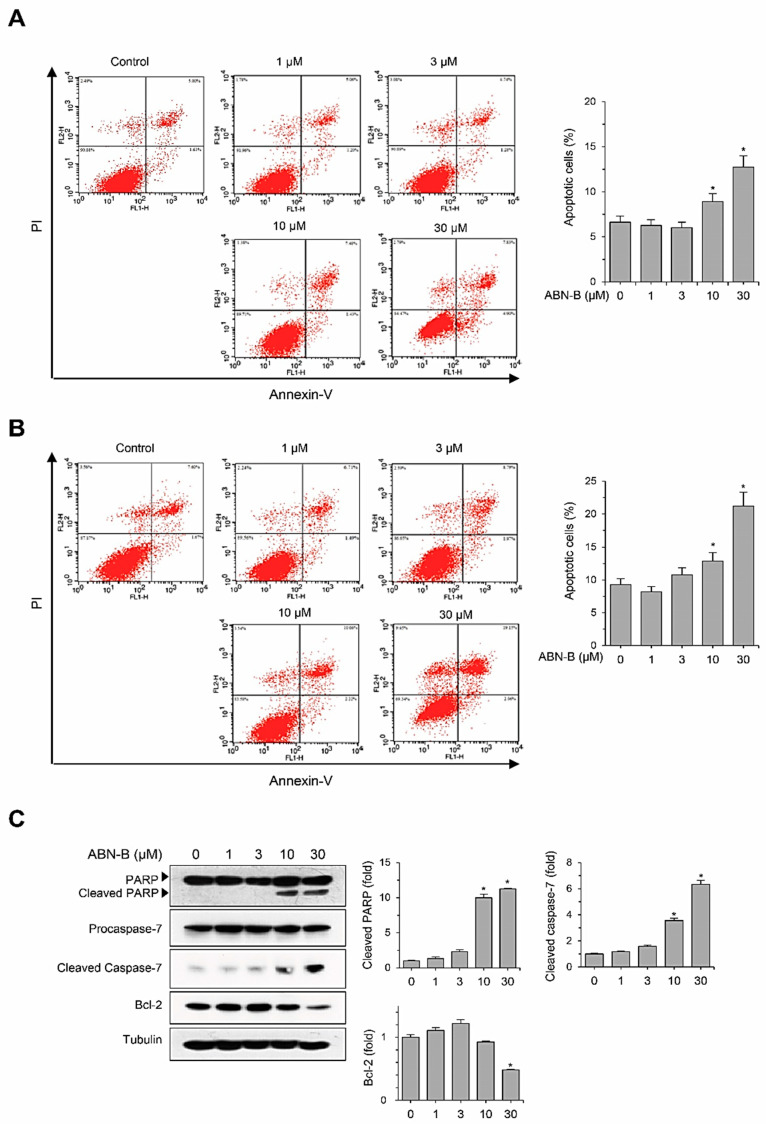
ABN-B induced apoptosis in A549 cells. (**A**,**B**) A549 cells were treated with the indicated concentrations of ABN-B for 24 h (**A**) or 48 h (**B**), subsequently stained with annexin V-FITC and PI, followed by analysis using a flow cytometry. Data represent the percentage of both early and late apoptotic cells and are expressed as the mean ± SEM of three independent experiments (* *p* < 0.01 compared with vehicle-treated control). (**C**) A549 cells were treated with indicated concentrations of ABN-B for 48 h. Subsequently, whole cell lysates were prepared, and we performed Western blot analysis with the indicated antibodies. Graphs represent densitometry analyses (* *p* < 0.01 compared with vehicle treated control, *n =* 3).

**Figure 4 ijms-21-09502-f004:**
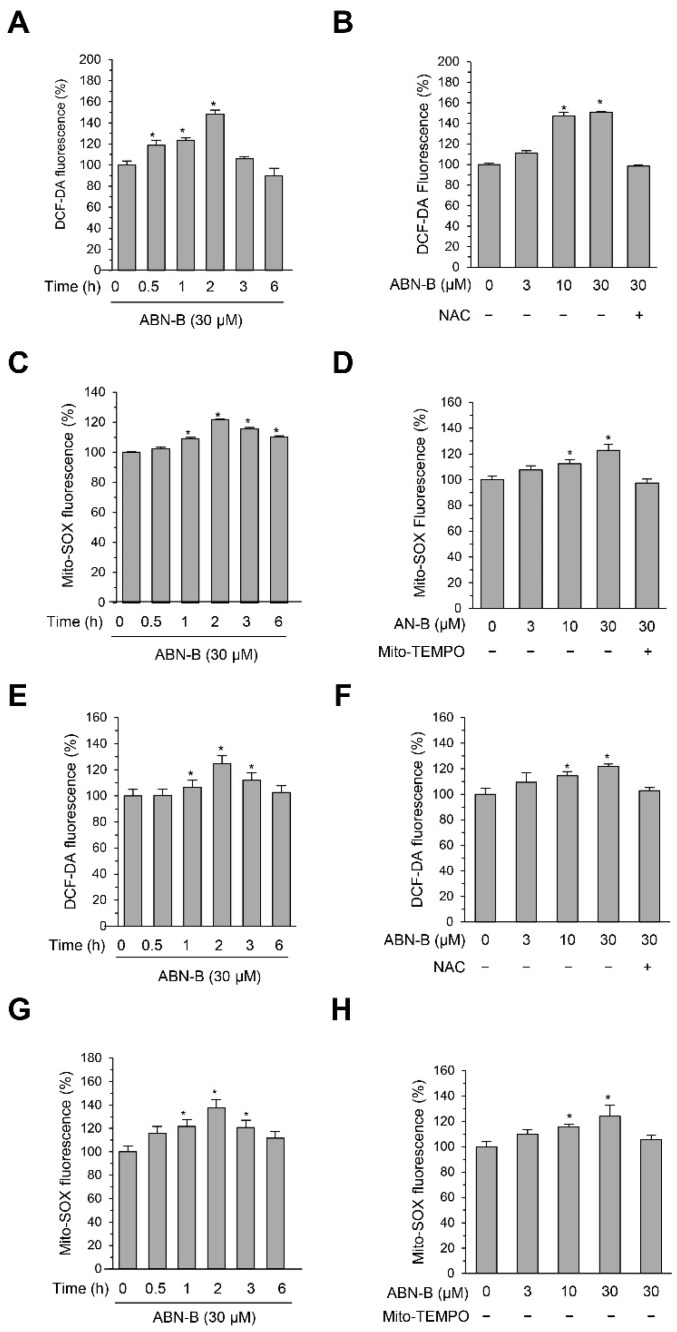
ABN-B induced ROS production in A549 and H1975 cells. (**A**,**E**) A549 (**A**) and NCI-H1975 (**E**) cells were treated with ABN-B (30 μM) for the indicated periods of time. DCF-DA was used to assess intracellular ROS generation and DCF-DA fluorescence was measured by flow cytometry. Data are mean ± SEM (* *p* < 0.01 compared with vehicle treated control, *n* = 3). (**B**,**F**) A549 (**B**) and NCI-H1975 (**E**) cells were treated with the indicated concentrations of ABN-B for 2 h with or without NAC (100 μM). DCF-DA fluorescence was measured by flow cytometry. Data are mean ± SEM (* *p* < 0.01 compared with vehicle treated control, *n* = 3). (**C**,**G**) A549 (**C**) and NCI-H1975 (**G**) cells were treated with ABN-B (30 μM) for the indicated periods of time. Mito-SOX was used to assess mitochondrial ROS generation and Mito-SOX fluorescence was measured by flow cytometry. Data are mean ± SEM (* *p* < 0.01 compared with vehicle treated control, *n* = 3). (**D**,**H**) A549 (**D**) and NCI-H1975 (**H**) cells were treated with the indicated concentrations of ABN-B for 2 h with or without Mito-TEMPO (50 μM). Mito-SOX fluorescence was measured by flow cytometry. Data are mean ± SEM (* *p* < 0.01 compared with vehicle treated control, *n* = 3).

**Figure 5 ijms-21-09502-f005:**
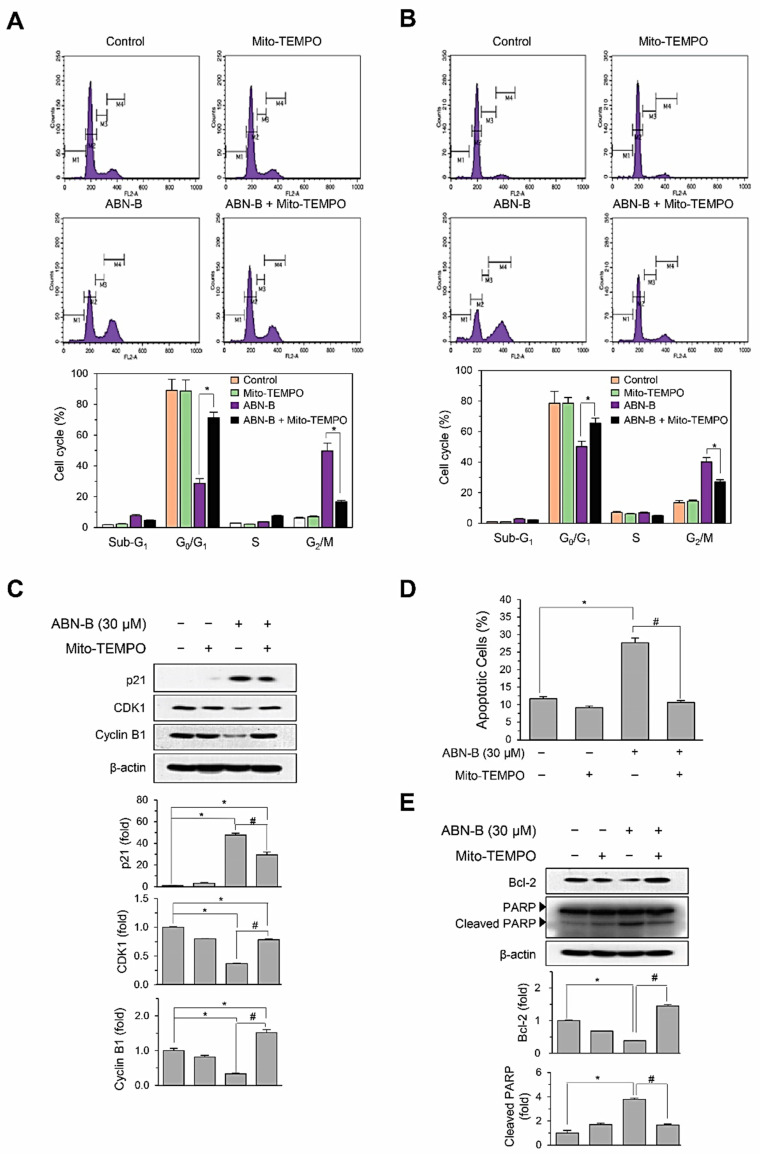
Mito-TEMPO attenuated cell cycle arrest at G_2_/M and apoptosis induced by ABN-B (**A**,**B**) A549 (**A**) and NCI-H1975 (**B**) cells were treated with of ABN-B (30 µM) with or without mito-TEMPO (50 µM) for 48 h, subsequently stained with PI, followed by analysis using a flow cytometry. Data are expressed as the mean ± SE of three independent experiments (* *p* < 0.01 compared with ABN-B-only treated group). (**C**) A549 cells were treated with of ABN-B (30 µM) with or without mito-TEMPO (50 µM) for 48 h. Subsequently, whole cell lysates were prepared, and performed Western blot analysis with the indicated antibodies. Graphs represent densitometry analyses (* *p* < 0.01 compared with vehicle-treated control; ^#^
*p* < 0.01 compared with ABN-B-only treated group). (**D**) A549 cells were treated with of ABN-B (30 µM) with or without mito-TEMPO (50 µM) for 48 h, subsequently stained with annexin V-FITC and PI, followed by analysis using a flow cytometry. Data are expressed as the mean of three independent experiments (* *p* < 0.01 compared with vehicle-treated control; ^#^
*p* < 0.01 compared with ABN-B-only treated group). (**E**) A549 cells were treated with of ABN-B (30 µM) with or without mito-TEMPO (50 µM) for 48 h. Subsequently, whole cell lysates were prepared, and performed Western blot analysis with the indicated antibodies. Graphs represent densitometry analyses (* *p* < 0.01 compared with vehicle-treated control; ^#^
*p* < 0.01 compared with ABN-B-only treated group).

**Figure 6 ijms-21-09502-f006:**
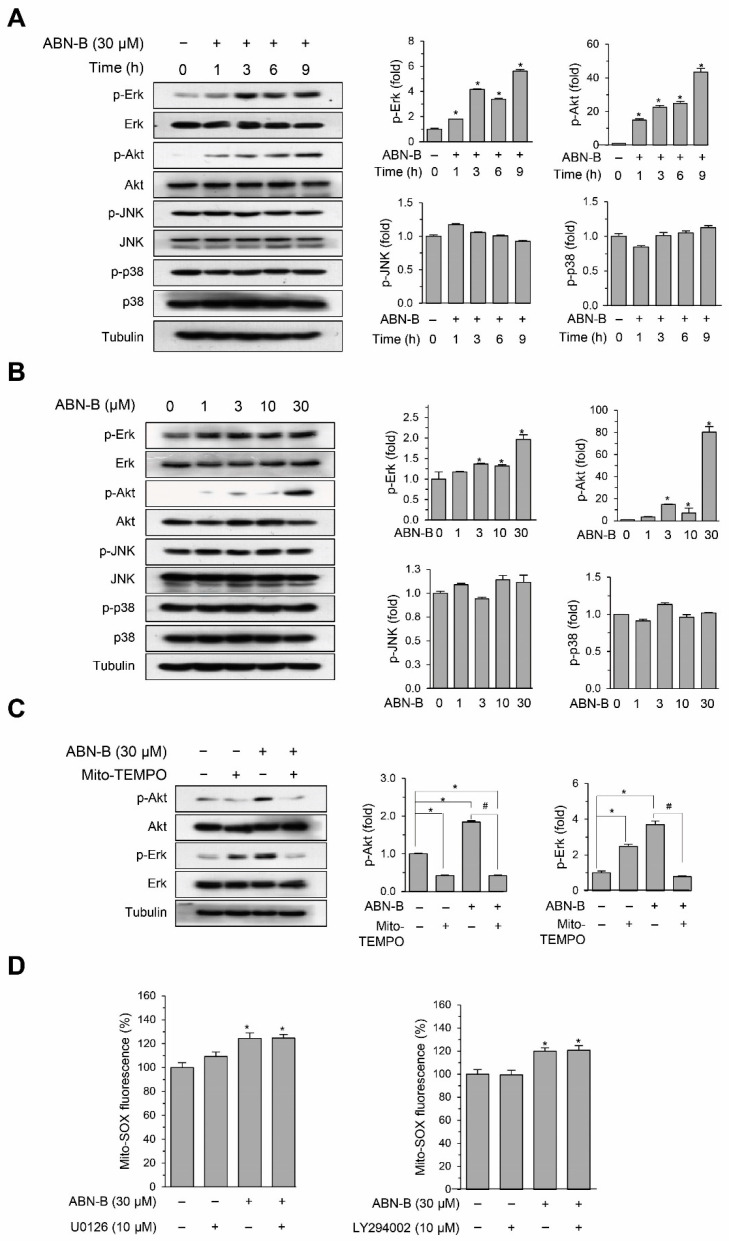
Mito-TEMPO attenuated ABN-B-induced phosphorylation of Akt and Erk1/2 in A549 cells. (**A**,**B**) A549 cells were with ABN-B (30 µM) for the indicated periods of time (**A**) or the indicated concentrations of ABN-B for 9 h (**B**). Subsequently, whole cell lysates were prepared, and we performed Western blot analysis with the indicated antibodies. Graphs represent densitometry analyses (* *p* < 0.01 compared with vehicle treated control, *n =* 3). (**C**) A549 cells were treated with of ABN-B (30 µM) with or without mito-TEMPO (50 µM) for 9 h. Subsequently, whole cell lysates were prepared, and we performed Western blot analysis with the indicated antibodies. Graphs represent densitometry analyses (* *p* < 0.01 compared with vehicle-treated control; ^#^
*p* < 0.01 compared with ABN-B-only treated group). (**D**) A549 cells were treated with ABN-B (30 μM) in the presence of U0126 or LY294002 for 2 h. Mito-SOX was used to assess mitochondrial ROS generation and Mito-SOX fluorescence was measured by flow cytometry. Data are mean ± SEM (* *p* < 0.01 compared with vehicle treated control, *n* = 3).

**Figure 7 ijms-21-09502-f007:**
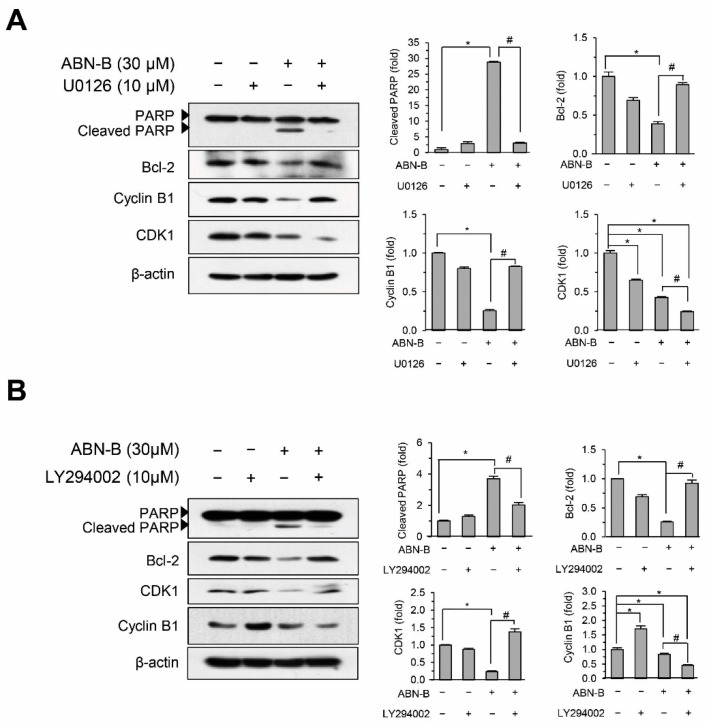
Inhibition of Erk1/2 and Akt attenuated apoptosis and cell cycle arrest at G_2_/M induced by ABN-B in A549 cells. (**A**,**B**) A549 cells were with ABN-B (30 µM) in the presence of U0126 (**A**) or LY294002 (**B**) for 48 h (**B**). Subsequently, whole cell lysates were prepared, and performed Western blot analysis with the indicated antibodies. Graphs represent densitometry analyses (* *p* < 0.01 compared with vehicle-treated control; ^#^
*p* < 0.01 compared with ABN-B-only treated group).

**Figure 8 ijms-21-09502-f008:**
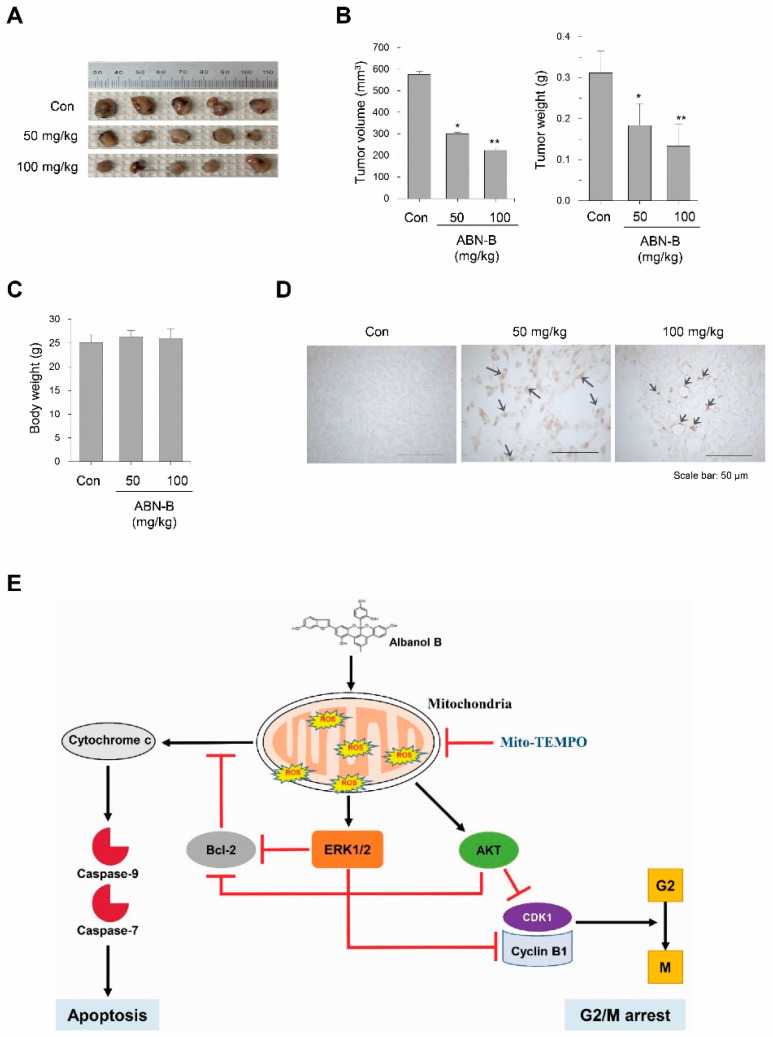
ABN-B suppressed tumor growth in Ex-3LL tumor-bearing mice. Ex-3LL tumor-bearing mouse were randomly divided into three groups: vehicle treatment group (Con), ABN-B 50 mg/kg treatment group, and ABN-B 100 mg/kg treatment group. The mice were sacrificed 21 days after treatment and tumors were removed. (**A**) Photographs of all tumors harvested. (**B**,**C**) Tumor volume (**B**) and tumor weight (**C**) of each treatment group (* *p* < 0.05 and ** *p* < 0.01 compared with vehicle treated control). (**D**) In situ apoptosis was detected TUNEL staining of tumor tissues sections from each treatment group (original magnification 400×). (**E**) A schematic diagram of action mechanism of ABN-B.

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
