# Peer review of "Albanol B from Mulberries Exerts Anti-Cancer Effect through Mitochondria ROS Production in Lung Cancer Cells and Suppresses In Vivo Tumor Growth"

_ijms, 2020, doi:10.3390/ijms21249502_

Round 1

Reviewer 1 Report

The paper “Albanol B from mulberries exerts anti-cancer effect through mitochondria ROS production in lung cancer cells” by Phan and collaborators studies the anti-cancer activity of albanol B from Morus alba in in vitro and in vivo lung cancer models, investigating the type of cell death, whether cell cycle was affected and the role of ROS production, as well as the factors implied.

The study is very rich and properly performed. However, the paper needs some improvements to be more readable and easier to be understood. For this reason, I believe that the paper will be suitable for publication after addressing different points:

1) In the title there is no mention of the in vivo experiment, hence it seems too narrow, in my point of view.

2) In the Results, paragraph 2.1 line 87, Authors performed the colony assay only with A549 and NCI-H1975 cells, why only these two?

3) Authors are warmly suggested to make all the figures much clearer since they turn out to be quite confusing. For example, in Figure 1, Authors can juxtapose images A and B while adding 24h and 48h on top to be more impactful.

4) For both intracellular and mitochondrial ROS evaluation, was NAC or Mito-TEMPO added contemporarily to ABN-B or before it? This is not clear either from the legend or from the methods.

5) The colony formation assay is written in a very confusing manner. How it is possible to have 500 cells in a 12-well plate? Can Authors cite the reference from which they drew this peculiar protocol?

6) Why Authors chose the intraperitoneally route in the in vivo study? Is there any report of pharmacokinetics of ABN-B?

Author Response

Reviewer 1:

The paper “Albanol B from mulberries exerts anti-cancer effect through mitochondria ROS production in lung cancer cells” by Phan and collaborators studies the anti-cancer activity of albanol B from Morus alba in in vitro and in vivo lung cancer models, investigating the type of cell death, whether cell cycle was affected and the role of ROS production, as well as the factors implied.

The study is very rich and properly performed. However, the paper needs some improvements to be more readable and easier to be understood. For this reason, I believe that the paper will be suitable for publication after addressing different points:

1) In the title there is no mention of the in vivo experiment, hence it seems too narrow, in my point of view.

RE]

According to the reviewer’s suggestion, we would kindly like to change the title to “Albanol B from mulberries exerts anti-cancer effect through mitochondria ROS production in lung cancer cells and suppresses in vivo tumor growth”. We changed the title in this revised version of manuscript.

2) In the Results, paragraph 2.1 line 87, Authors performed the colony assay only with A549 and NCI-H1975 cells, why only these two?

RE]

MTT assay results showed that albanol B could effectively suppress the proliferation of all four lung cell lines in a concentration- and time-dependent manner. Non-small cell lung cancer (NSCLC) comprise approximately 80–85% of all lung cancers (our reference # 30). Thus, we selected two NSCLS cancer cell lines, H1975 and A549 cells, for further investigation of anticancer effect of ABN-B. As suggested by reviewer, we described this issue accordingly in this revised version of manuscript (Please see from line 89 to 93 in the revised version).

3) Authors are warmly suggested to make all the figures much clearer since they turn out to be quite confusing. For example, in Figure 1, Authors can juxtapose images A and B while adding 24h and 48h on top to be more impactful.

RE]

As suggested, we changed the details in figures (Please see Figures 1, 2 and 5 in the revised version of manuscript)

4) For both intracellular and mitochondrial ROS evaluation, was NAC or Mito-TEMPO added contemporarily to ABN-B or before it? This is not clear either from the legend or from the methods.

RE]

As suggested by reviewer, we described this issue in “Materials and Methods” section (please see from line # 417 to 421).

5) The colony formation assay is written in a very confusing manner. How it is possible to have 500 cells in a 12-well plate? Can Authors cite the reference from which they drew this peculiar protocol?

RE]

The colony formation assay is a widely used method to study the number and size of cancer cell colonies that remain after treatment of anti-cancer agents and serves as a measure for the anti-proliferative effect of these agents (Nat. Protoc. 1, 2315–2319, 2006; PLoS ONE 9(3): e92444, 2014. doi:10.1371/journal.pone.0092444). As suggested, we cited a reference in the revised version of manuscript (please see our reference # 49 in the revised version)

6) Why Authors chose the intraperitoneally route in the in vivo study? Is there any report of pharmacokinetics of ABN-B?

RE]

Injection of substances into the peritoneal cavity is a common technique in laboratory rodents (Please see, J Am Assoc Lab Anim Sci. 2011 Sep;50(5):600-13), and has been commonly used for evaluation of the pharmacological activities of compounds. Although intraperitoneal delivery is considered a parenteral route of administration, the pharmacokinetics of substances administered intraperitoneally are more similar to those seen after oral administration, because the primary route of absorption is into the mesenteric vessels, which drain into the portal vein and pass through the liver (J Am Assoc Lab Anim Sci. 2011 Sep;50(5):600-13).

To our knowledge, there is no any report about pharmacokinetics of ABN-B. We are also interested in this issue. Such studies are currently beginning with the aid of collaborators; however, inclusion of prospective studies would not be possible in a reasonable time frame. Please understand our situation.

Reviewer 2 Report

The paper “Albanol B from mulberries exerts anti-cancer effect through mitochondria ROS production in lung cancer cells” by Phan and co-workers investigates the anti-proliferative activity of the arylbenzofuran albanol B from Morus alba in four lung cancer cell lines as well as in an in vivo model of lung carcinoma, deeply studying the mechanisms of action.

Overall, the study is well-set and rich of experimental procedures. The logic thread is nicely followed and the appropriateness of the whole work makes it really sound. Nevertheless, the paper sometimes may sound reiterative and confusing. Therefore, it can be accepted for publication after addressing few major points:

1) In the Results, paragraph 2.1 line 82, Authors claimed that the survival rate of the tested cancer cell lines decreased in both concentration- and time-dependent manner; however, looking at the figure, this is not true for all four cell lines. Therefore, Authors are suggested to be more precise.

2) In the Results, paragraph 2.1 line 85, Authors referred to etoposide without showing the results in figure; I think that they should present these results to help the reader understand better.

3) In Figure 2, as for all the figures, Authors should make it clearer. Moreover, how is it possible that the 24h treatment of albanol B 30 µM (figure 2 D) gave stronger effect respect to the same concentration in the same cell line but at a longer timing? Why Authors chose only A549 cells to perform Western blot analyses, if cell cycle ones had nice results for both cell lines?

4) In Figure 3, other than making it clearer, what did Authors intend with those bars for apoptotic cells? Did they mean early and late apoptotic taken together? Why only A549, if even NCI-H1975 gave good results for cell cycle analysis? Please, be more precise.

5) In Figures 5, 6 and 7, the symbols for statistics should differ whether referring to vehicle control or mitochondrial scavenger one, otherwise it is really difficult to understand the significance.

6) In Material and Methods, Authors cited a work for the extraction of albanol B from M. alba, even though this paper does not explain any extraction procedure at all. Authors should fix this issue.

7) When performing Western blot, is common sense that phosphorylated proteins need different attention (i.e. phosphatase inhibitors, sodium fluoride in milk, etc.). Authors should mention it in the proper paragraph.

8) Given the important number of results Authors presented in this paper, the Discussion section, though having a proper scheme, should be enriched to support better Authors’ results.

Author Response

Reviewer 2: Comments and Suggestions for Authors

The paper “Albanol B from mulberries exerts anti-cancer effect through mitochondria ROS production in lung cancer cells” by Phan and co-workers investigates the anti-proliferative activity of the arylbenzofuran albanol B from Morus alba in four lung cancer cell lines as well as in an in vivo model of lung carcinoma, deeply studying the mechanisms of action.

Overall, the study is well-set and rich of experimental procedures. The logic thread is nicely followed and the appropriateness of the whole work makes it really sound. Nevertheless, the paper sometimes may sound reiterative and confusing. Therefore, it can be accepted for publication after addressing few major points:

1) In the Results, paragraph 2.1 line 82, Authors claimed that the survival rate of the tested cancer cell lines decreased in both concentration- and time-dependent manner; however, looking at the figure, this is not true for all four cell lines. Therefore, Authors are suggested to be more precise.

RE]

As suggested, we described more precisely in the Result section (Please see from line # 81 to 93 in the revised version).

2) In the Results, paragraph 2.1 line 85, Authors referred to etoposide without showing the results in figure; I think that they should present these results to help the reader understand better.

RE]

As suggested, we presented this data in Supplementary Figure 1 in the revised version of manuscript.

3) In Figure 2, as for all the figures, Authors should make it clearer. Moreover, how is it possible that the 24h treatment of albanol B 30 µM (figure 2 D) gave stronger effect respect to the same concentration in the same cell line but at a longer timing? Why Authors chose only A549 cells to perform Western blot analyses, if cell cycle ones had nice results for both cell lines?

RE]

1) As suggested by the reviewer, we corrected Figure 2 more clearer.

2) We understand the reviewer’s concern. In fact, we performed Western blot analysis regarding to the data for Figure 2C and 2D at least twice. Moreover, we confirmed the expression level of cyclin B1 at 30 µM of ABN-B after 48 h treatment. Constantly, treatment of ABN-B at almost completely suppressed the expression of CDK1 and cyclin B1 for 24 or 48 h incubation, as shown in Figure 2C and 2D. We replaced Western blot for cyclin B1 in Figure 2C with a new blot, and attached this original image of cyclin B1 in Figure 2C.

3) A549 cells were more sensitive to ABN-B for inhibiting BrdU incorporation and inducing cell-cycle arrest than H1299 cell lines. Thus, we chose A549 for further studies.

4) In Figure 3, other than making it clearer, what did Authors intend with those bars for apoptotic cells? Did they mean early and late apoptotic taken together? Why only A549, if even NCI-H1975 gave good results for cell cycle analysis? Please, be more precise.

Re]

The bar graphs in Figure 3A and 3B represent the total percentage of both early and late apoptotic cells. We described this issue in the revised version of manuscript (Please see from line # 146 to 147).

As pointed-out by the reviewer, we understand the reviewer’s concern. As shown in Figures 1C, 2A, and 2B, A549 cells were more sensitive to ABN-B for inhibiting BrdU incorporation and inducing cell-cycle arrest than H1299 cell lines. Thus, we chose A549 for further studies.

5) In Figures 5, 6 and 7, the symbols for statistics should differ whether referring to vehicle control or mitochondrial scavenger one, otherwise it is really difficult to understand the significance.

RE]

As pointed-out, we corrected these errors in this revised version of manuscript (please see Figure legend for Figures 5-7 in the revised version).

6) In Material and Methods, Authors cited a work for the extraction of albanol B from M. alba, even though this paper does not explain any extraction procedure at all. Authors should fix this issue.

RE]

As suggested by the reviewer, we described the extraction procedure in Materials and Methods section (please see line # 364 to 374 in the revised version).

7) When performing Western blot, is common sense that phosphorylated proteins need different attention (i.e. phosphatase inhibitors, sodium fluoride in milk, etc.). Authors should mention it in the proper paragraph.

Re: We mentioned the using of sodium orthovanadate in the Materials and methods section, according to reviewer’s comment (please see line # 402 in the revised version).

8) Given the important number of results Authors presented in this paper, the Discussion section, though having a proper scheme, should be enriched to support better Authors’ results.

RE]

As suggested, we added a scheme for mode of action mechanism(s) of ABN-B in Figure 8E and cited accordingly (Please see Figure 8 and line # 279).

Round 2

Reviewer 1 Report

Authors have thoroughly answered to all my questions.

Author Response

NA

Reviewer 2 Report

The paper “Albanol B from mulberries exerts anti-cancer effect through mitochondria ROS production in lung cancer cells” by Phan and co-workers investigates the anti-proliferative activity of the arylbenzofuran albanol B from Morus alba in four lung cancer cell lines as well as in an in vivo model of lung carcinoma, deeply studying the mechanisms of action.

Overall, the study is well-set and rich of experimental procedures. The logic thread is nicely followed and the appropriateness of the whole work makes it really sound. Nevertheless, the paper sometimes may sound reiterative and confusing. Therefore, it can be accepted for publication after addressing few major points:

1) In the Results, paragraph 2.1 line 82, Authors claimed that the survival rate of the tested cancer cell lines decreased in both concentration- and time-dependent manner; however, looking at the figure, this is not true for all four cell lines. Therefore, Authors are suggested to be more precise.

RE]

As suggested, we described more precisely in the Result section (Please see from line # 81 to 93 in the revised version).

okay

2) In the Results, paragraph 2.1 line 85, Authors referred to etoposide without showing the results in figure; I think that they should present these results to help the reader understand better.

RE]

As suggested, we presented this data in Supplementary Figure 1 in the revised version of manuscript.

okay

3) In Figure 2, as for all the figures, Authors should make it clearer. Moreover, how is it possible that the 24h treatment of albanol B 30 µM (figure 2 D) gave stronger effect respect to the same concentration in the same cell line but at a longer timing? Why Authors chose only A549 cells to perform Western blot analyses, if cell cycle ones had nice results for both cell lines?

RE]

1) As suggested by the reviewer, we corrected Figure 2 more clearer.

okay

2) We understand the reviewer’s concern. In fact, we performed Western blot analysis regarding to the data for Figure 2C and 2D at least twice. Moreover, we confirmed the expression level of cyclin B1 at 30 µM of ABN-B after 48 h treatment. Constantly, treatment of ABN-B at almost completely suppressed the expression of CDK1 and cyclin B1 for 24 or 48 h incubation, as shown in Figure 2C and 2D. We replaced Western blot for cyclin B1 in Figure 2C with a new blot, and attached this original image of cyclin B1 in Figure 2C.

okay

3) A549 cells were more sensitive to ABN-B for inhibiting BrdU incorporation and inducing cell-cycle arrest than H1299 cell lines. Thus, we chose A549 for further studies.

 okay, but it is better to explain this also in the text, in my opinion.

4) In Figure 3, other than making it clearer, what did Authors intend with those bars for apoptotic cells? Did they mean early and late apoptotic taken together? Why only A549, if even NCI-H1975 gave good results for cell cycle analysis? Please, be more precise.

Re]

The bar graphs in Figure 3A and 3B represent the total percentage of both early and late apoptotic cells. We described this issue in the revised version of manuscript (Please see from line # 146 to 147).

okay

As pointed-out by the reviewer, we understand the reviewer’s concern. As shown in Figures 1C, 2A, and 2B, A549 cells were more sensitive to ABN-B for inhibiting BrdU incorporation and inducing cell-cycle arrest than H1299 cell lines. Thus, we chose A549 for further studies.

 okay, but, as said before, but it is better to explain this also in the text, at least once.

5) In Figures 5, 6 and 7, the symbols for statistics should differ whether referring to vehicle control or mitochondrial scavenger one, otherwise it is really difficult to understand the significance.

RE]

As pointed-out, we corrected these errors in this revised version of manuscript (please see Figure legend for Figures 5-7 in the revised version).

 okay

6) In Material and Methods, Authors cited a work for the extraction of albanol B from M. alba, even though this paper does not explain any extraction procedure at all. Authors should fix this issue.

RE]

As suggested by the reviewer, we described the extraction procedure in Materials and Methods section (please see line # 364 to 374 in the revised version).

The paragraph is very clear and focused. Yet, the problem still remains because reference 24 has not any extraction method in it, so it should be removed.

7) When performing Western blot, is common sense that phosphorylated proteins need different attention (i.e. phosphatase inhibitors, sodium fluoride in milk, etc.). Authors should mention it in the proper paragraph.

Re: We mentioned the using of sodium orthovanadate in the Materials and methods section, according to reviewer’s comment (please see line # 402 in the revised version).

 okay

8) Given the important number of results Authors presented in this paper, the Discussion section, though having a proper scheme, should be enriched to support better Authors’ results.

RE]

As suggested, we added a scheme for mode of action mechanism(s) of ABN-B in Figure 8E and cited accordingly (Please see Figure 8 and line # 279).

Actually, I meant a different thing when suggesting to improve the discussion. Nevertheless, the chosen approach to gather all the results in a resuming image is clever and lead to the outcome I expected.

Author Response

Reviewer 2 requests to address two points as follows.

1) Why Authors chose only A549 cells to perform Western blot analyses, if cell cycle ones had nice results for both cell lines? it is better to explain this also in the text, in my opinion.

RE]

As suggested, we described this issue in this revised version (Please see from line# 125 to 127).

2) The reference #24 has not any extraction method in it, so it should be removed.

RE]

We deleted this reference in M & M section (Please see line# 366).